# MoReact: Generating Reactive Motion from Textual Descriptions

**Xiyan Xu**                                                    *xiyanxu2@illinois.edu*
*University of Illinois Urbana-Champaign*

**Sirui Xu**                                                    *siruixu2@illinois.edu*
*University of Illinois Urbana-Champaign*

**Yu-Xiong Wang**                                               *yxw@illinois.edu*
*University of Illinois Urbana-Champaign*

**Liang-Yan Gui**                                               *lgui@illinois.edu*
*University of Illinois Urbana-Champaign*

**Reviewed on OpenReview:** *https://openreview.net/forum?id=4zuT73heqm*

## Abstract

Modeling and generating human reactions poses a significant challenge with broad applications for computer vision and human-computer interaction. Existing methods either treat multiple individuals as a single entity, directly generating interactions, or rely solely on one person's motion to generate the other's reaction, failing to integrate the rich semantic information that underpins human interactions. Yet, these methods often fall short in adaptive responsiveness, *i.e.*, the ability to accurately respond to diverse and dynamic interaction scenarios. Recognizing this gap, our work introduces an approach tailored to address the limitations of existing models by focusing on text-driven human reaction generation. Our model specifically generates realistic motion sequences for individuals that responding to the other's actions based on a descriptive text of the interaction scenario. The goal is to produce motion sequences that not only complement the opponent's movements but also semantically fit the described interactions. To achieve this, we present MoReact, a diffusion-based method designed to disentangle the generation of global trajectories and local motions sequentially. This approach stems from the observation that generating global trajectories first is crucial for guiding local motion, ensuring better alignment with given action and text. Furthermore, we introduce a novel interaction loss to enhance the realism of generated close interactions. Our experiments, utilizing data adapted from a two-person motion dataset, demonstrate the efficacy of our approach for this novel task, which is capable of producing realistic, diverse, and controllable reactions that not only closely match the movements of the counterpart but also adhere to the textual guidance. Please find our webpage at `https://xiyan-xu.github.io/MoReactWebPage/`.

## 1 Introduction

Humans are able to naturally interact with one another, adopting appropriate actions based on their intentions and the movements of others. For instance, when someone extends their hand, we understand it as an invitation for a handshake, allowing us to react accordingly – either by engaging in the handshake or choosing to ignore and walk away. Since interactions are a fundamental aspect of the real world, reproducing this behavior pattern in virtual characters holds profound implications for various fields, such as robotics, animation, VR/AR, and healthcare. It enhances realism and functionality across these fields, enriching user experiences and interactions with technology.

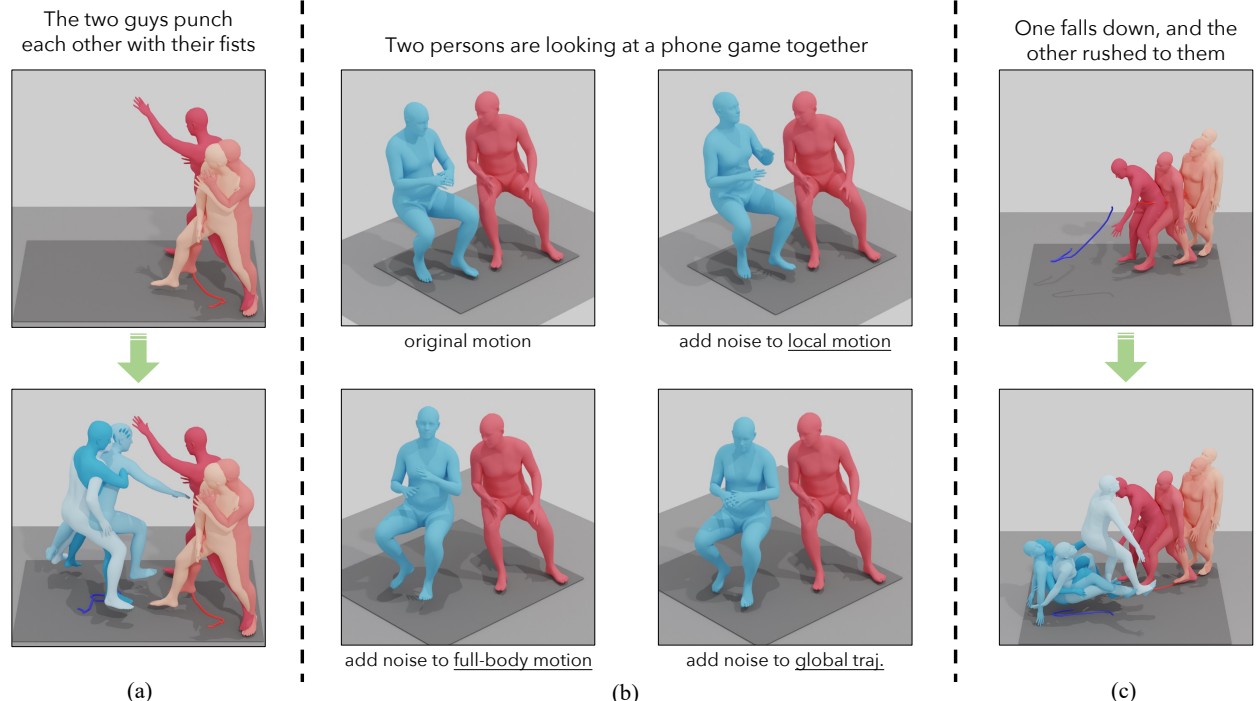

Figure 1: (**a**) Our model, MoReact, learns to generate lifelike reactions, represented by the blue mesh, based on the textual description and the actor's motion, represented by the red mesh. (**b**) As an important motivating analysis for developing our approach, we introduce noise of the same scale to local motion, full-body motion, and global trajectory, respectively. The results indicate that the precision of the global trajectory has a greater impact on the perceptual realism of the reaction. (**c**) We demonstrate global trajectory's significant influence on the motion's semantic information in certain scenarios, such as fall actions.

However, achieving realistic generation of human reactions is a significant challenge. Current approaches to Human-Human Interaction (HHI) generation have not yet fully captured the complexity of human reactions, particularly in generating adaptive reactions to diverse and dynamic social interactions. Current studies, while striving to generate reactive motions for a variety of actions (Ghosh et al., 2024; Xu et al., 2024a;b; Liu et al., 2024; Tan et al., 2025; Liu et al., 2023; Cen et al., 2025; Cong et al., 2025), typically lack integration of the rich semantic information that underlies human interactions, such as the intent conveyed in text descriptions.

Addressing these limitations, we propose a novel approach that focuses on enhancing text-driven human reaction generation. Specifically, we aim to more accurately generate one's (named *reactor*) reaction based on both the motion of another individual (named *actor*) and a corresponding textual description, as shown in Fig. 1(a). This approach not only seeks to align generated reactions with physical movements but also with the semantic context provided by text, presenting unique challenges in maintaining both alignment and authenticity of interactions.

Naïvely adapting existing text-to-motion (Shafir et al., 2023) or text-to-interaction (Liang et al., 2024) models fails to generate high-quality reactions based on text descriptions, often resulting in artifacts like interaction misalignment. The underlying reason is that these methods struggle to adequately model the *relationship between global trajectory and local motion*. To begin with, most existing methods for two-person interaction generation either overly focus on local motion or treat these two equally. However, our analysis indicates that the global trajectory serves as the foundation for both local motion and interaction realism. Incorrect global trajectory makes it difficult for the local motion to align with the action and text description, having a more detrimental impact on interaction realism than incorrect local motion. As shown in Fig. 1(b),

minor deviations in local motion have little impact on interaction quality, while adding the same scale of deviations in global motion leads to the reaction being misaligned, though the single-human motion can still be reasonable if the interaction is not considered. Furthermore, most existing methods neglect the substantial influence of the global trajectory on the semantics of local motion; instead, they persist in exploring a broad spectrum of potential reactions, rather than focusing on a more confined and plausible subspace delineated by the global trajectory. For example, as shown in Fig. 1(c), the descent of the global trajectory to a lower height suggests that the reactor ought to fall, rather than stand or walk, to achieve a lower position.

Building upon these observations and insights, we propose a novel framework, MoReact, to tackle the text-driven human interaction generation task. MoReact incorporates two primary components: (1) *Trajectory Diffusion Module*: This diffusion-based generator predicts the reactor's global trajectory based on the text description of the interaction and the actor's motion. (2) *Full-Body Motion Diffusion Module*: This diffusion-based generator incorporates text, actor's motion, and reactor's trajectory from the Trajectory Diffusion Module as inputs to synthesize the reactor's full-body reaction. It further enhances the reaction's realism by incorporating the reactor's trajectory information into the denoising process through an inpainting mechanism. By decoupling the reaction generation into two phases, MoReact places a strong emphasis on generating accurate global trajectories. The infilling of accurate global trajectories generated in the first stage to the full-body motion diffusion model further guides the local motion generation to a specific, refined plausible subspace. Doing so guarantees the semantic integrity of local motion, contributing to the overall realism of the generated reaction.

Within this two-stage modeling framework, we further introduce the interaction loss – a specialized loss function tailored to characterize reactions and interactions. Rather than solely depending on the reconstruction loss, which targets global absolute coordinates or local joint features, we highlight the significance of *relative* motion in interactions. Specifically, we focus on the motion of the reactor's joints in relation to the actor's joint movements, crafting a weighted interaction graph to emphasize the influential relative motion. For instance, in scenarios where two joints are close to each other, *e.g.*, two hands in a handshake interaction, we specifically utilize the interaction loss to promote such contact to be accurately represented.

Based on these features, MoReact generates realistic and high-quality reactions that follow the text guidance faithfully while also syncing seamlessly with the actor's motion. Furthermore, MoReact demonstrates remarkable control capacity – it is capable of synthesizing varied reactions to the actor's same motion based on different textual descriptions, and conversely, generating diverse reactions to actors' various motions guided by the same text, as shown in Fig. 4.

In summary, our contributions are: (**a**) We develop a novel two-stage diffusion-based modeling framework, MoReact, which incorporates inherent motion pattern in reactions and synthesizes global trajectory and full-body motion in a sequential manner, ensuring the generated reactions are of high quality. (**b**) We introduce a novel interaction loss that enhances the modeling of relative joint motions in interactions, utilizing a weighted interaction graph to accurately model close interactions such as handshakes. (**c**) We conduct extensive experimental evaluation that demonstrates MoReact's remarkable flexibility and controllability in the reaction generation process, enabling the synthesis of diverse reactions based on varying conditions. Comprehensive analyses are provided to validate our design decisions, showcasing the efficacy of MoReact in creating realistic reactions.

## 2 Related Work

**Human Motion Generative Models.** Generative models have achieved remarkable advancements in synthesizing human motion based on various inputs, such as action labels (Guo et al., 2020; Petrovich et al., 2021; Lee et al., 2023; Athanasiou et al., 2022), audio signals (Li et al., 2022; Tseng et al., 2023; Li et al., 2021; Yi et al., 2023; Zhou & Wang, 2023), prior motions (Xu et al., 2022; Barquero et al., 2023; Chen et al., 2023a), movement trajectories (Kaufmann et al., 2020; Karunratanakul et al., 2023; Rempe et al., 2023; Xie et al., 2024), the surrounding environment (Cao et al., 2020; Hassan et al., 2021; Wang et al., 2021b;c; 2022a;b; Huang et al., 2023; Zhao et al., 2022; 2023; Tendulkar et al., 2023; Zhang et al., 2024a), and including methods that generate motions without any specific conditions (Raab et al., 2023a). Particularly in text-based human motion generation (Petrovich et al., 2023; Guo et al., 2022b; Petrovich

et al., 2022; Tevet et al., 2023; Chen et al., 2023b; Zhang et al., 2022; Jiang et al., 2023; Zhang et al., 2024b; 2023b; Tevet et al., 2022; Ahuja & Morency, 2019; Guo et al., 2022a; Kim et al., 2023; Lu et al., 2023; Raab et al., 2023b; Zhang et al., 2023a; Shafir et al., 2023; Dabral et al., 2023; Zhang et al., 2023c; Wei et al., 2023; Zhang et al., 2024c; Athanasiou et al., 2023; Kong et al., 2023), significant progress has been made through the integration of diffusion models (Sohl-Dickstein et al., 2015; Song et al., 2020; Ho et al., 2020). A notable advantage of diffusion models is their capacity to iteratively refine the generation process by reintroducing available information, thus tailoring the outcomes to specific conditions. An instance of this is PhysDiff (Yuan et al., 2023), which incorporates a physics-based motion imitation policy into the diffusion process to produce physically realistic motion. GMD (Karunratanakul et al., 2023) is capable of generating human motions tailored to specific goals, such as following a trajectory or achieving certain keyframes. This is also accomplished by strategically incorporating information into the diffusion process. In our work, we adopt a simple but effective inpainting mechanism to incorporate global trajectory information into the full-body motion generation process, ensuring the full-body motion aligns coherently with the intended trajectory.

**Interactive Motion Synthesis.** Interactive motion generation takes into account both human movements and the dynamics of interactive entities, including objects and other humans. The goal is to generate motions that are both realistic and appropriately responsive to the interactive context. For example, human-object interaction generation (Xu et al., 2023b; Li et al., 2024; Diller & Dai, 2024; Peng et al., 2025; Starke et al., 2019; Wang et al., 2023; Merel et al., 2020; Hassan et al., 2023; Bae et al., 2023; Corona et al., 2020) considers the dynamics of both humans and objects. Human-human interaction generation (Wang et al., 2021a; Xu et al., 2023a;c; Adeli et al., 2020; 2021; Guo et al., 2022c; Tanke et al., 2023a; Zhu et al., 2023; Tanke et al., 2023b; Peng et al., 2023; Liu et al., 2023; Cai et al., 2024; Wang et al., 2024; Liang et al., 2024; Shafir et al., 2023; Rempe et al., 2023; Xu et al., 2024a;b; Liu et al., 2024) involves motions from two or more persons. For methods aimed at modeling interactions in crowds with numerous participants, there is a notable emphasis on global trajectories synthesis. For instance, DuMMF (Xu et al., 2023c) separates the modeling of local and global representations in social interactions, placing constraints on global motion. Trace and Pace (Rempe et al., 2023) present a trajectory-guided diffusion model that allows for the manipulation of trajectories while considering the context of the surrounding environment. However, in two-person interaction generation, the crucial role of global trajectory modeling has been overlooked. In our work, we generate global trajectory and local motion sequentially, allowing the global trajectory to guide the generation of local motion. Despite the advancements in this field, these attempts at modeling multi-person interactions do not adequately capture individual reactions to others. For example, they often struggle to synchronize or align with rapid and varied movements. In contrast, some recent works (Ghosh et al., 2024; Xu et al., 2024b; Liu et al., 2024; Tan et al., 2025; Liu et al., 2023; Cen et al., 2025; Cong et al., 2025) focus on reactive motion generation, emphasizing the spatio-temporal coherence between the actor's and reactor's movements. However, these models are limited as they generate reactions either solely based on the actor's motion or by combining the actor's motion with an action label, which fail to capture the full diversity of potential reactions. To address this, we enhance the reactive motion generation by incorporating textual guidance.

## 3 Methodology

**Overview.** In this section, we begin by formally defining the text-driven reaction generation task. We then introduce the overall architecture of MoReact in Sec. 3.1. In Sec. 3.2 and Sec. 3.3, we delve into the model's design and the detailed configurations of MoReact's training process. Finally, in Sec. 3.4 we introduce the inpainting mechanism we used during the inference stage.

**Task Definition.** Given a motion sequence of the actor and a sentence describing the interaction between the actor and the reactor, our goal is to generate the reactor's motion that not only harmoniously coordinates with the actor but also aligns coherently with the textual description. In this context, **actor** refers to the character with known motion, while **reactor** refers to the character for whom we aim to synthesize motion, *a.k.a.*, reaction.

The reactor's motion sequence with $T$ frames, is represented as $\boldsymbol{x} = [\boldsymbol{x}^1, \boldsymbol{x}^2, \ldots, \boldsymbol{x}^T]$, where each $\boldsymbol{x}^i \in \mathbb{R}^{263}$ contain the pose information of the reactor at the $i$-th frame in an adapted HumanML3D (Guo et al., 2022a) representation. Specifically, each pose $\boldsymbol{x}^i$ consists of both global trajectory information $\boldsymbol{g}^i \in \mathbb{R}^4$ and local

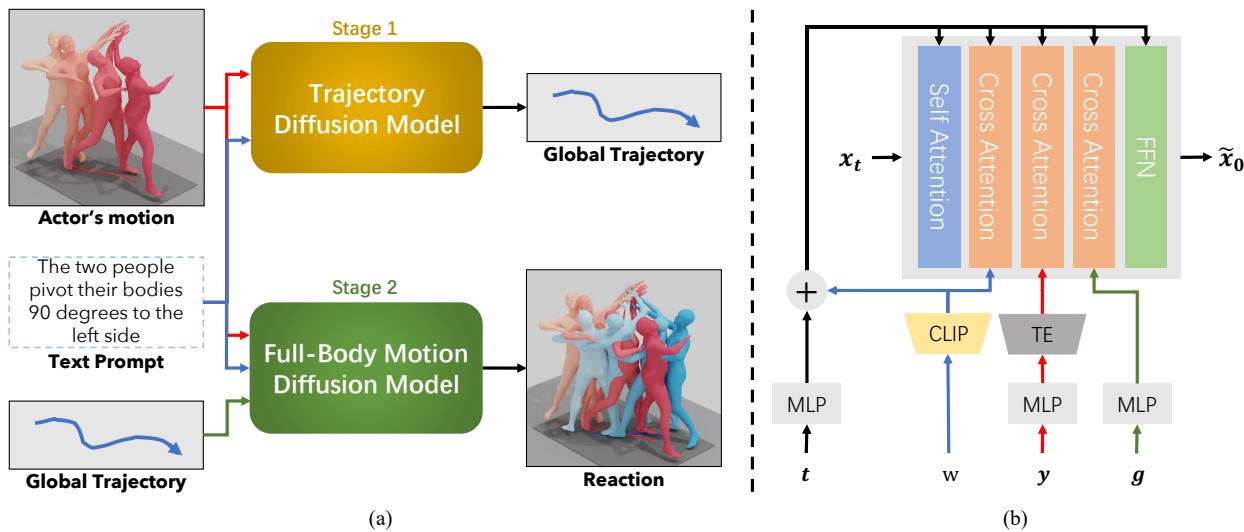

Figure 2: **Overview of MoReact**. (**a**) Our approach to text-driven reaction generation employs a two-stage framework. First, we employ a trajectory diffusion model to generate the global trajectory of the reactor, based on the actor's full-body motion and the text description. Subsequently, we apply a full-body motion diffusion model to generate the reactor's full-body motion, based on the actor's full-body motion, the text description, as well as the synthesized reactor's trajectory. (**b**) Our full-body motion diffusion model is built upon a transformer-based architecture, where the 'TE' in the figure denotes a Transformer Encoder. The trajectory diffusion model mirrors this architecture but omits the cross-attention layer that integrates global trajectory into the generative process.

pose information $l^i \in \mathbb{R}^{259}$. Global trajectory information $g^i$ consists of root orientation along the y-axis and 3D translation in the global coordinate, while local pose information $l^i$ includes joint position, joint velocity, joint rotation, and foot contact in the reactor's local coordinate. Similarly, we use $y = [y^1, y^2, \ldots, y^T]$ to denote the motion sequence of the actor.

We use $w = [w_1, w_2, \cdots, w_n]$ to denote the $n$-words textual description of the interaction between the actor and the reactor. Our goal is to model the conditional probability distribution $p(x|y, w)$, from which we can then generate the reactor's motion sequence through sampling.

## 3.1 Framework Overview

As shown in Fig. 2(a), we decouple the process for text-driven reaction generation into two sequential stages, consisting of two diffusion modules, *i.e.*, the Trajectory Diffusion Module and the Full-Body Motion Diffusion Module. In the first stage, we use the Trajectory Diffusion Module to generate the global trajectory of the reactor, informed by the motion of the actor and the text description. Following this, in the second stage, we utilize the Full-Body Motion Diffusion Module to generate the full-body motion of the reactor, based on the actor's motion, the previously generated reactor's global trajectory, and the text description. During the inference stage, we adopt an inpainting mechanism during the denoising process to ensure the final full-body motion faithfully adheres to the intended trajectory pre-generated by Trajectory Diffusion Module.

## 3.2 Text-Driven Reaction Generation

**Trajectory Generation.** Based on our key observation that global motion significantly outweighs local motion in generating reactions, our approach prioritizes the reactor's overall trajectory by generating it at the beginning, and using it as the condition for the full-body generation. We use $G_{\text{traj}}$ to denote our diffusion model for this generation. More specifically, $G_{\text{traj}}$ takes time step $t$, a noised global trajectory $g_t$, the actor's motion sequence $y$, and the text condition description $w$ as input. We observe that instead of

estimating the clean denoised signal $\tilde{\boldsymbol{g}}_0$, estimating the noise $\tilde{\boldsymbol{\epsilon}}_t = G_{\text{traj}}(t, \boldsymbol{g}_t, \boldsymbol{y}, w)$ added in the forward diffusion, and recovering $\tilde{\boldsymbol{g}}_0$ with $\tilde{\boldsymbol{\epsilon}}_t$ leads to better trajectory generation, which is consistent with the finding in GMD (Karunratanakul et al., 2023).

**Full-Body Motion Generation.** In this stage, our goal is to generate realistic, coherent, and semantically aligned full-body reactions based on a synthesized global trajectory. Similarly, we use $G_{\text{full}}$ to denote our diffusion model for the reactor's full-body motion generator. We input time step $t$, a noised full-body motion $\boldsymbol{x}_t$, the actor's motion $\boldsymbol{y}$, the text condition description $w$, as well as the synthesized reactor's trajectory $\boldsymbol{g}$ into the model $G_{\text{full}}$. Following MDM (Tevet et al., 2023), the model $G_{\text{full}}$ is designed to directly predict the clean motion $\tilde{\boldsymbol{x}}_0$ as the output. Note that during the training phase, we utilize the reactor's ground truth trajectory $\boldsymbol{g}$ as the input for the model, while we directly use the synthesized trajectory during the inference phase.

**Network Architectures.** Fig. 2(b) illustrates the architecture of $G_{\text{full}}$. Given a noisy motion $\boldsymbol{x}_t$, our model feeds it into a transformer-style architecture and obtains the denoised motion $\tilde{\boldsymbol{x}}_0$. This architecture consists of self-attention blocks, cross-attention blocks, and feed-forward networks. The cross-attention layers integrate the motion feature with the features extracted from $w$, $\boldsymbol{y}$, and $\boldsymbol{g}$.

### 3.3 Loss Formulations with Training Details

We train the trajectory generation diffusion model, $G_{\text{traj}}$, and the full-body motion generation model, $G_{\text{full}}$, *independently*. And they can be seamlessly combined during inference. The loss formulations and the training strategies are elaborated in the following.

**Trajectory Generation.** In the trajectory generation stage, we use a single reconstruction loss term to train our model. Formally, the loss function $L_{\text{traj}}$ is defined as:

$$L_{\text{traj}} = \|\boldsymbol{\epsilon} - \tilde{\boldsymbol{\epsilon}}\|_2^2, \tag{1}$$

where $\tilde{\boldsymbol{\epsilon}} = G_{\text{traj}}(t, \boldsymbol{g}_t, \boldsymbol{y}, w)$ represents the estimated noise.

**Full-Body Motion Generation.** At the stage of full-body motion generation, the loss function consists of three terms: the reconstruction loss $L_{\text{R}}$, the kinematic loss $L_{\text{K}}$, and the interaction loss $L_{\text{I}}$. We define the training objective $L_{\text{full}}$ in this stage as:

$$L_{\text{full}} = \lambda_{\text{R}} L_{\text{R}} + \lambda_{\text{K}} L_{\text{K}} + \lambda_{\text{I}} L_{\text{I}}. \tag{2}$$

Here, $\lambda_{\text{R}}, \lambda_{\text{K}}, \lambda_{\text{I}}$ are the weights assigned to $L_{\text{R}}, L_{\text{K}}, L_{\text{I}}$, respectively.

(**i**) Similar to the reconstruction loss used in the trajectory generation stage, we define $L_{\text{R}} = \|\boldsymbol{x}_0 - \tilde{\boldsymbol{x}}_0\|_2^2$ as the reconstruction loss here. However, solely relying on the reconstruction loss does not necessarily lead to realistic reaction.

(**ii**) To address the problem such as foot sliding or jittering artifacts, following (Tevet et al., 2023; Liang et al., 2024; Ghosh et al., 2024), we use a kinematic loss term $L_{\text{K}}$. To specify, $L_{\text{K}}$ consists of 4 subterms, which are $L_{\text{K}}^{\text{foot}}, L_{\text{K}}^{\text{vel}}, L_{\text{K}}^{\text{rot}}, L_{\text{K}}^{\text{traj}}$, representing foot skating loss, velocity loss, global rotation loss, and global trajectory loss, respectively. With the weights $\lambda_{\text{foot}}, \lambda_{\text{vel}}, \lambda_{\text{rot}}, \lambda_{\text{traj}}$, $L_{\text{K}}$ can be formulated as:

$$L_{\text{K}} = \lambda_{\text{foot}} L_{\text{K}}^{\text{foot}} + \lambda_{\text{vel}} L_{\text{K}}^{\text{vel}} + \lambda_{\text{rot}} L_{\text{K}}^{\text{rot}} + \lambda_{\text{traj}} L_{\text{K}}^{\text{traj}}. \tag{3}$$

For more details on the formulation of these subterms, please refer to Appendix D.

(**iii**) We introduce an interaction loss, $L_{\text{I}}$, that emphasizes the spatial relationships within interactions. Inspired by (Zhang et al., 2023d), this approach models human interactions as an interaction graph where each joint of the actor and reactor serves as a node, and edges represent joint pairs. To compute $L_{\text{I}}$, we design a weighting function that emphasizes pairs of joints that are closer together, deeming pairs that are farther apart as less critical for interaction.

Specifically, the interaction loss $L_{\text{I}}$ consists of two terms: position interaction loss $L_{\text{I}}^p$ and velocity interaction loss $L_{\text{I}}^v$, as shown in Eq. 4, where $\lambda_p$ and $\lambda_v$ are corresponding weights:

$$L_{\text{I}} = \lambda_p L_{\text{I}}^p + \lambda_v L_{\text{I}}^v. \tag{4}$$

To compute $L_I^p$, we first use forward kinematics to calculate the joint coordinates of the generated reaction and the actor's motion in the global coordinate, denoted as $\tilde{\boldsymbol{P}}_x$ and $\boldsymbol{P}_y \in \mathbb{R}^{J \times T \times 3}$, where $T$ is the motion length and $J$ is the number of joints. Next, we calculate the interaction graph $\tilde{\boldsymbol{M}}_p \in \mathbb{R}^{J \times J \times T \times 3}$, where $\tilde{\boldsymbol{M}}_p[i,j] = \boldsymbol{P}_y[j] - \tilde{\boldsymbol{P}}_x[i]$, representing the difference in coordinates for each frame between the $i$-th joint of the reactor and the $j$-th joint of the actor. We also compute distance graph $\tilde{\boldsymbol{D}}_p \in \mathbb{R}^{J \times J \times T}$, where $\tilde{\boldsymbol{D}}_p[i,j] = \|\tilde{\boldsymbol{M}}_p[i,j]\|_2$, representing the distance between joint pair $(i,j)$. Similarly, we calculate $\boldsymbol{M}_p$ and $\boldsymbol{D}_p$ of the ground truth reactor and the actor. The position interaction loss $L_I^p$ can be formulated as:

$$L_I^p = \frac{1}{|S|} \sum_{(i,j,k) \in S} \boldsymbol{W}_p[i,j,k] \|\tilde{\boldsymbol{M}}_p[i,j,k] - \boldsymbol{M}_p[i,j,k]\|_2^2, \tag{5}$$

where $S = \{(i,j,k) | \boldsymbol{D}_p[i,j,k] \leq c\}$ is the set of joints-frame pairs that the distance between the $i$-th joint of the reactor and the $j$-th joint of the actor is below a threshold $c$ in the ground truth distance graph. The weighted term $W_p \in \mathbb{R}^{J \times J \times T}$ is defined as:

$$\boldsymbol{W}_p = (\sigma(\tilde{\boldsymbol{D}}_p) + \sigma(\boldsymbol{D}_p))(\phi(\tilde{\boldsymbol{D}}_p) + \phi(\boldsymbol{D}_p)), \tag{6}$$

where $\sigma(\boldsymbol{D})[i,j,k] = \frac{exp(\boldsymbol{D}[i,j,k])}{\sum_{x,y} exp(\boldsymbol{D}[x,y,k])}$ is a softmax function along the joint pair axis, and $\phi(\boldsymbol{D})[i,j,k] = \frac{1}{\boldsymbol{D}[i,j,k]}$. The intuition behind $S$ and weighted terms $W_p$ is that distant joint pairs are less important for interaction, while closer joint pairs are more critical. We can compute the velocity interaction loss $L_I^v$ similarly. Please refer to Appendix D for details.

As highlighted in (Yuan et al., 2023; Liang et al., 2024; Xu et al., 2023b), we find that the denoising motion in the initial denoising phase does not have reasonable kinematic and interaction properties. Based on this observation, it is not reasonable to apply kinematic and interaction losses if the diffusion time step $t$ is large. Thus, we adopt a thresholding scheme among loss functions during the training of our full-body motion generator. Specifically, we set a threshold $\bar{t}$ and only apply the kinematic loss and the interaction loss if $t$ is no larger than $\bar{t}$. Therefore, the final form of $L_{\text{full}}$ can be written as:

$$L_{\text{full}} = \lambda_R L_R + I(t \leq \bar{t})(\lambda_I L_I + \lambda_K L_K). \tag{7}$$

### 3.4 Inference

In the inference stage, we incorporate an inpainting mechanism into the denoising process, similar to the motion infilling scheme in (Tevet et al., 2023; Shafir et al., 2023; Raab et al., 2023b). Specifically, at each time step $t$, after estimating the clean sample $\tilde{\boldsymbol{x}}_0$, we fuse the $\tilde{\boldsymbol{x}}_0$ with the previously generated global trajectory $\boldsymbol{g}$ from the first stage to form $\hat{\boldsymbol{x}}_0$. This process can be formulated as:

$$\hat{\boldsymbol{x}}_0 = (1 - M) \odot \tilde{\boldsymbol{x}}_0 + M \odot \boldsymbol{g}, \tag{8}$$

where $M$ represents the mask for the dimensions that describe the global trajectory information in the motion feature $\boldsymbol{x}$ and $\odot$ is the Hadamard product. The modified result, $\hat{\boldsymbol{x}}_0$, is then used to sample $\boldsymbol{x}_{t-1}$. Through this inpainting mechanism, we continuously incorporate the known global trajectory $\boldsymbol{g}$ into the denoising process, ensuring the final full-body motion faithfully adheres to $\boldsymbol{g}$.

## 4 Experiments

In this section, we begin by introducing the experimental setup of our work, which includes evaluation metrics, baseline settings, and implementation details in Sec. 4.1. Subsequently, we present the quantitative results of our method in Sec. 4.2, followed by the qualitative results in Sec. 4.3. Finally, in Sec. 4.4, we discuss the ablation study conducted on our model.

### 4.1 Experimental Setup

**Dataset.** We conduct our evaluation on the InterHuman (Liang et al., 2024) and CHI3D (Fieraru et al., 2020) datasets. InterHuman features 6,022 motion sequences across various interaction categories, annotated

with 16,756 unique descriptions. We use the official training and testing split as specified in InterHuman. To demonstrate the generalizability of MoReact, we also evaluate its performance on the action-driven reaction generation task using the CHI3D dataset, which contains 376 interaction sequences with action labels. We split this dataset into training and testing sets at a 2:1 ratio.

**Evaluation Metrics.** We adopt the evaluation metrics used in prior works (Liang et al., 2024; Xu et al., 2024b) for our quantitative analysis, evaluating the actor and reactor's motion as a whole. Specifically, in the experiments on the InterHuman dataset, **R-Precision** measures the relevance of the generated interaction to the provided text description. The Fréchet Inception Distance (**FID**) evaluates the realism of the generated reaction relative to the actor's motion. Additionally, we use Multi-Modality Distance (**MM Dist**) to assess the alignment between the text and the generated interaction in a shared latent space. We also measure **Diversity** to examine the variation in the generated motions. For the experiments on the CHI3D dataset, we further evaluate the generated interactions by measuring their action recognition **Accuracy** using a pretrained action classifier.

**Baseline.** Given that text-driven human reaction generation is a novel task, *there is no existing work or code publicly available that directly serves as a baseline for comparison*. To facilitate comparison, in our evaluations, we modify InterGen (Liang et al., 2024) to incorporate an inpainting mechanism during the inference stage, as described in Sec. 3.4. This modification ensures that InterGen takes the actor's motion as a condition to generate the reactor's motion accordingly. We also adapted MDM (**?**), a widely used method in text-driven human motion generation, to suit the text-driven reaction generation task. For more details, please refer to the Appendix D.

Table 1: **Quantitative Comparison on InterHuman and CHI3D.** $\pm$ represents the 95% confidence interval, and $\rightarrow$ indicates that values that closer to the Real are better. * indicates the model is evaluated without motion infilling mechanism.

| Methods | InterHuman | | | | CHI3D | | |
|---|---|---|---|---|---|---|---|
| | 3-Precision↑ | FID↓ | MM Dist↓ | Diversity→ | Accuracy↑ | FID↓ | Diversity→ |
| Real | $0.704^{\pm0.005}$ | $0.206^{\pm0.009}$ | $3.784^{\pm0.001}$ | $7.799^{\pm0.031}$ | $0.604^{\pm0.005}$ | $0.084^{\pm0.005}$ | $3.995^{\pm0.056}$ |
| MDM | $0.532^{\pm0.006}$ | $3.763^{\pm0.056}$ | $3.844^{\pm0.001}$ | $7.751^{\pm0.021}$ | $0.496^{\pm0.011}$ | $13.850^{\pm0.375}$ | $\mathbf{3.997}^{\pm0.056}$ |
| MDM-GRU | $\mathbf{0.640}^{\pm0.006}$ | $12.758^{\pm0.158}$ | $\mathbf{3.812}^{\pm0.001}$ | $7.640^{\pm0.028}$ | $0.345^{\pm0.011}$ | $39.280^{\pm1.397}$ | $4.097^{\pm0.089}$ |
| InterGen | $0.631^{\pm0.005}$ | $7.207^{\pm0.114}$ | $\mathbf{3.812}^{\pm0.001}$ | $7.692^{\pm0.038}$ | $0.531^{\pm0.017}$ | $46.531^{\pm0.699}$ | $4.082^{\pm0.077}$ |
| InterGen* | $0.614^{\pm0.008}$ | $7.576^{\pm0.178}$ | $3.821^{\pm0.002}$ | $7.860^{\pm0.051}$ | $0.661^{\pm0.005}$ | $14.772^{\pm0.448}$ | $\mathbf{3.962}^{\pm0.058}$ |
| MoReact | $0.615^{\pm0.007}$ | $\mathbf{2.412}^{\pm0.050}$ | $3.813^{\pm0.002}$ | $\mathbf{7.775}^{\pm0.046}$ | $\mathbf{0.687}^{\pm0.014}$ | $\mathbf{10.801}^{\pm0.313}$ | $3.582^{\pm0.063}$ |

Table 2: **Ablation Studies on InterHuman dataset.** The results demonstrate the effectiveness of kinematic loss $L_K$, interaction loss $L_I$, thresholding scheme and two-stage framework.

| Methods | 3-Precision↑ | FID↓ | MM Dist↓ | Diversity→ |
|---|---|---|---|---|
| Real | $0.704^{\pm0.005}$ | $0.206^{\pm0.009}$ | $3.784^{\pm0.001}$ | $7.799^{\pm0.031}$ |
| MoReact(w/o $L_K, L_I$) | $\mathbf{0.623}^{\pm0.007}$ | $3.164^{\pm0.062}$ | $\mathbf{3.808}^{\pm0.001}$ | $7.719^{\pm0.031}$ |
| MoReact(w/o $L_I$) | $0.594^{\pm0.005}$ | $2.456^{\pm0.030}$ | $3.816^{\pm0.002}$ | $7.832^{\pm0.037}$ |
| MoReact(w/o $L_K$) | $0.618^{\pm0.008}$ | $2.673^{\pm0.020}$ | $3.810^{\pm0.002}$ | $7.784^{\pm0.036}$ |
| MoReact(w/o threshold scheme) | $0.613^{\pm0.008}$ | $3.403^{\pm0.044}$ | $3.816^{\pm0.002}$ | $\mathbf{7.796}^{\pm0.031}$ |
| MoReact(single-stage) | $0.591^{\pm0.007}$ | $2.776^{\pm0.032}$ | $3.822^{\pm0.002}$ | $7.761^{\pm0.051}$ |
| MoReact | $0.615^{\pm0.007}$ | $\mathbf{2.412}^{\pm0.050}$ | $3.813^{\pm0.002}$ | $7.775^{\pm0.046}$ |

**Implementation Details.** As a pre-processing step, we normalize the actor's motion by relocating it to the origin and rotating it to make the actor facing z+ axis. Subsequent transformations are applied to the reactor to maintain the spatial relationship between the actor and reactor unchanged. Similar to previous work (Tevet et al., 2023; Zhang et al., 2022), we use a pretrained and frozen CLIP (Radford et al., 2021) model to encode text prompts into text features, while the rest of MoReact is trained from scratch. The trajectory generation model is trained for 1,200 epochs, and the full-body motion generation model is trained for 2,000 epochs. We train both models using a learning rate of $lr = 1e - 4$ and the AdamW optimizer,

with a batch size of 32. Our models are implemented in PyTorch and trained on two NVIDIA A40 GPUs. Following (Tevet et al., 2023), we adopt a classifier-free approach (Ho & Salimans, 2022) in the generation process. For evaluation, we adopt the MotionClip (Tevet et al., 2022) provided by InterGen (Liang et al., 2024) to evaluate our model's performance on InterHuman dataset. We follow the implementation of ST-GCN (Yan et al., 2018) to train the action recognition model and compute action-classification accuracy on CHI3D (Fieraru et al., 2020) dataset. Additional implementation details are provided in the Appendix D.

## 4.2 Quantitative Results

We compare MoReact with InterGen (Liang et al., 2024) and MDM (Tevet et al., 2023) on the test sets of the InterHuman and CHI3D datasets. For a fair comparison, we include the results of both adapted InterGen with actor motion infilling and the original InterGen without motion infilling. The main results are summarized in Table 1. Our approach, MoReact, outperforms the baselines across various metrics, particularly achieving significant improvements in interaction quality and alignment, as evidenced by a substantial reduction in FID. As described in Sec. 4.1, FID is the most critical metric for reflecting the quality of the generated reactions. The results also indicate that naively adapting existing text-interaction or text-motion models fails to generate satisfactory reactions. This underscores the need for new paradigms in text-driven reaction generation, implying the importance of our proposed method.

## 4.3 Qualitative Results

The qualitative comparisons between MoReact and baselines are demonstrated in Fig. 3. As shown in the first two rows, the baselines fail to generate realistic reactions based on the textual description and the actor's motion. Specifically, the reactions generated by InterGen and MDM do not faithfully align with the textual description and fail to coordinate harmoniously with the actor's motion. For instance, in Fig. 3(c), InterGen is unable to synthesize a motion that the reactor falls onto the ground as the text described. This exemplifies how the previously generated global trajectory can act as a constraint, narrowing the search space for full-body motion generation and resulting in interactions that better align with the text. Moreover, in Fig. 3(a), the reactor and actor overlap in the same space, leading to implausible body penetration artifacts. All visualization results from MDM fail to generate realistic reactions that align well with the actor's motion, even if the reaction itself is coherent with the text description. These issues highlight the importance of a plausible global trajectory for realistic interaction and demonstrate the effectiveness of MoReact.

Moreover, we demonstrate MoReact's capability for diverse control, as illustrated in Fig. 4. MoReact effectively synthesizes diverse reactions to the same motion of the actor based on differing textual descriptions. Conversely, it is also adept at generating varied reactions to different motions of actors when guided by the same text.

## 4.4 Ablation Studies

We conduct an ablation study on the InterHuman dataset to evaluate the effectiveness of our loss designs and two-stage framework, as shown in Table 2. The results indicate that the kinematic loss $L_K$, interaction loss $L_I$, and thresholding scheme all contribute to generating more realistic reactions and achieving lower FID scores. To validate the superiority of the two-stage framework, we develop a baseline model that operates in a single stage, directly generating full-body motion from the text description and the actor's motion. The quantitative results in Table 2 demonstrate that our two-stage model significantly outperforms the single-stage model across all metrics. Additional details of the ablation studies can be found in the Appendix E.

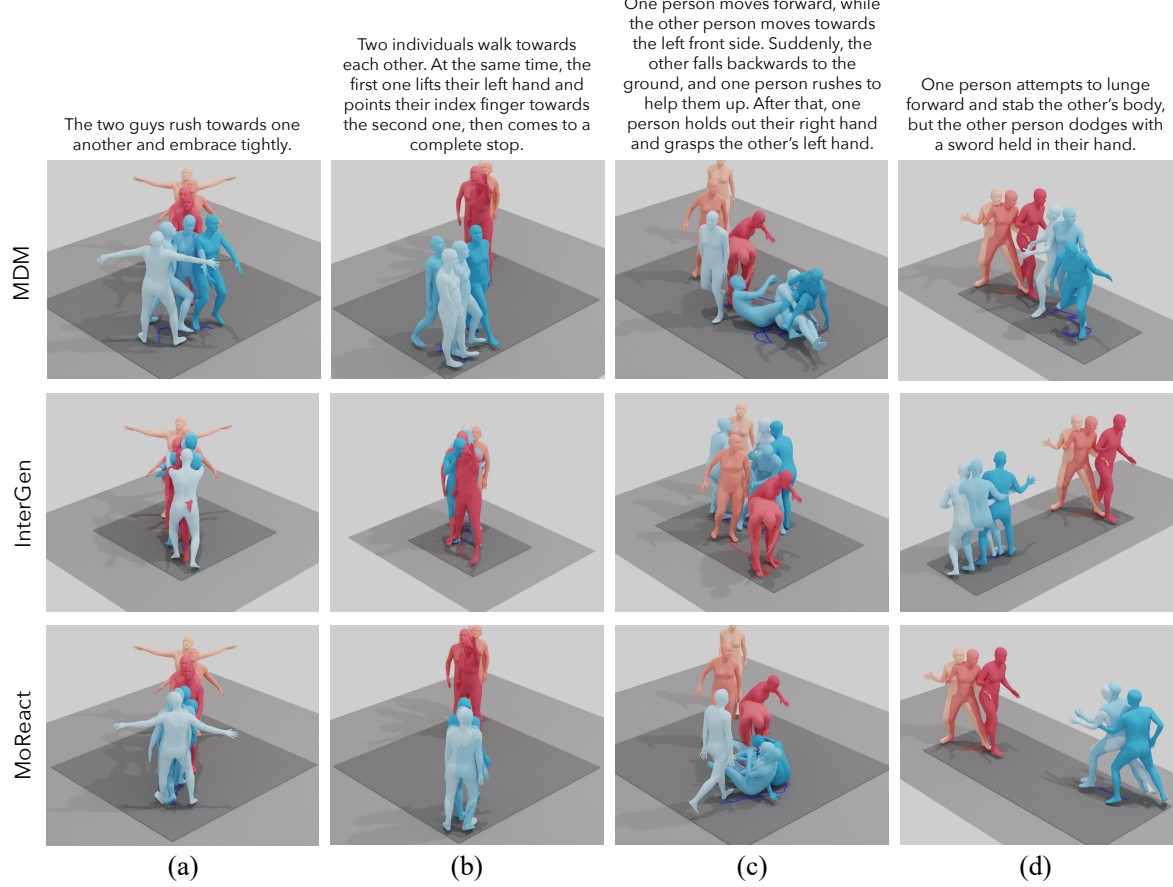

Figure 3: **Qualitative comparison.** We show that MoReact consistently generates more realistic reactions than MDM InterGen, avoiding issues such as body penetration (a)(b), text-motion mismatch (c), and interaction misalignment (d).

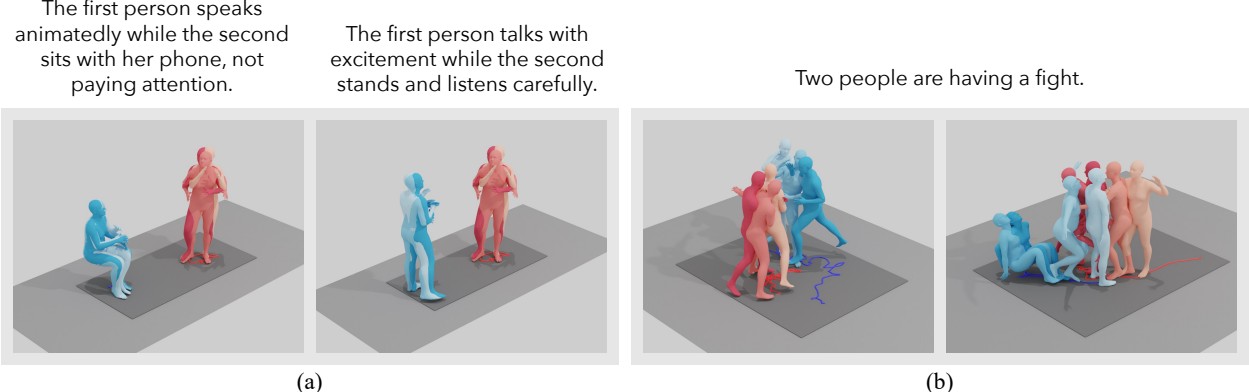

Figure 4: **Qualitative evaluation of the control capacity.**

## 5 Conclusions

In this work, we propose an innovative method, MoReact, to solve the text-driven human reaction generation. Utilizing a two-stage framework that sequentially synthesizes the global trajectory and full-body motion, MoReact effectively creates high-quality, realistic reactions. These reactions not only align accurately with the text but also harmonize with the actor's movements. Moreover, MoReact also demonstrates remarkable flexibility and controllability in the reaction generation process, enabling the synthesis of diverse reactions based on varying conditions. Experimental results demonstrate our methods' superiority over baselines and validate the efficacy of our model's design.

## Acknowledgments

This work was supported in part by NSF Grant 2106825, NIFA Award 2020-67021-32799, the Amazon-Illinois Center on AI for Interactive Conversational Experiences, the Toyota Research Institute, the IBM-Illinois Discovery Accelerator Institute, and Snap Inc. This work used computational resources, including the NCSA Delta and DeltaAI and the PTI Jetstream2 supercomputers through allocations CIS230012, CIS230013, and CIS240311 from the Advanced Cyberinfrastructure Coordination Ecosystem: Services & Support (ACCESS) program, as well as the TACC Frontera supercomputer, Amazon Web Services (AWS), and OpenAI API through the National Artificial Intelligence Research Resource (NAIRR) Pilot.

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

## A  Appendix Overview

In this appendix, we present further analyses, implementation details, and additional experimental results. Specifically: (1) We provide a detailed video demonstration in the supplementary materials, with a corresponding explanation provided in Sec. B. (2) We delve deeper into our key insight – global trajectory serves as the foundation for both local motion and interaction realism – and further validate this claim with quantitative experimental evidence in Sec. C. (3) Additional information regarding the implementation of MoReact and the baseline models is detailed in Sec. D. (4) Sec. E illustrates extra ablation studies to show the efficacy of MoReact's framework. (5) We discuss MoReact's limitations and social impacts in Sec. F.

## B  Visualization Demo

In the supplementary materials, we include a demo video that shows the visualization results associated with figures in the main paper. The video features: (1) visualizations of our main insights; (2) qualitative comparisons with baseline models; (3) showcases of MoReact's controllability on both text and motion; and (4) qualitative results of MoReact in action-driven reaction generation task on CHI3D dataset. Please watch the video for further results and details.

## C  A Further Investigation on Our Key Insight: The Central Role of Global Trajectory

As discussed and qualitatively validated in the main paper as well as in the demo video, a crucial insight of our work is that *the global trajectory serves as the foundation for both local motion and interaction realism. Incorrect global trajectory makes it difficult for the local motion to align with the action and text description, having a more detrimental impact on interaction realism than incorrect local motion.* Here to further validate this insight in a *quantitative* manner, we perform an experiment examining the impact of equivalent levels of noise on both global trajectory and local motion and their effects on the overall motion's realism.

In detail, for a reaction $\boldsymbol{x}_0$, we use a diffusion style forward process to apply a sequence of Gaussian noise additions to $x_0$ and obtain the noised full-body reactions $\boldsymbol{x}_1, \boldsymbol{x}_2, \cdots, \boldsymbol{x}_T$, where $\boldsymbol{x}_T \sim \mathcal{N}(\boldsymbol{0}, \boldsymbol{I})$. Based on the noised full-body reactions $\{\boldsymbol{x}_t\}_{t=1}^T$, we can obtain reactions with noised global trajectory $\{\boldsymbol{x}_t^g\}_{t=1}^T$ and reactions with noised local motion $\{\boldsymbol{x}_t^l\}_{t=1}^T$ with the following equations:

$$\boldsymbol{x}_t^g = (1 - M^g) \odot \boldsymbol{x}_0 + M^g \odot \boldsymbol{x}_t \tag{9}$$

$$\boldsymbol{x}_t^l = (1 - M^l) \odot \boldsymbol{x}_0 + M^l \odot \boldsymbol{x}_t, \tag{10}$$

where $M^g$, $M^l$ represent the masks for the dimensions that describe the global trajectory information and local motion information in the motion feature $\boldsymbol{x}$ respectively, and $\odot$ is the Hadamard product. For a set of time steps $\{t\}$, we compute $\boldsymbol{x}_t$, $\boldsymbol{x}_t^g$ and $\boldsymbol{x}_t^l$ for every reaction $\boldsymbol{x}$ in the test dataset. Subsequently, we evaluate the realism of interaction between the actor's motion and the noised reaction by calculating the Fréchet Inception Distance (**FID**) across the entire test dataset. The **FID** is computed by the MotionClip (Tevet et al., 2022) provided by InterGen (Liang et al., 2024). The results, depicted in Fig. C.1 and consistent with the demo video, show that adding noise to the global trajectory has a more detrimental effect on the realism of interactions compared with adding noise to the local motion, thus motivating the design of our MoReact framework.

## D  Implementation Details

**Formulation of Velocity Interaction Loss $L_1^v$.** Beyond the position interaction loss introduced in the main paper, we also employ a velocity interaction loss to enhance the model's ability to generate realistic close interactions. Similar to the computation of $L_1^p$, for $L_1^v$, we first compute $\tilde{\boldsymbol{V}}_x$ and $\boldsymbol{V}_y \in \mathbb{R}^{J \times (T-1) \times 3}$, representing the joint velocities of the reactor and the actor. We then calculate the velocity interaction graph $\tilde{\boldsymbol{M}}_v \in \mathbb{R}^{J \times J \times (T-1) \times 3}$, where $\tilde{\boldsymbol{M}}_v[i,j] = \boldsymbol{V}_y[j] - \tilde{\boldsymbol{V}}_x[i]$. We also calculate $\boldsymbol{M}_v$ for the ground truth reactor

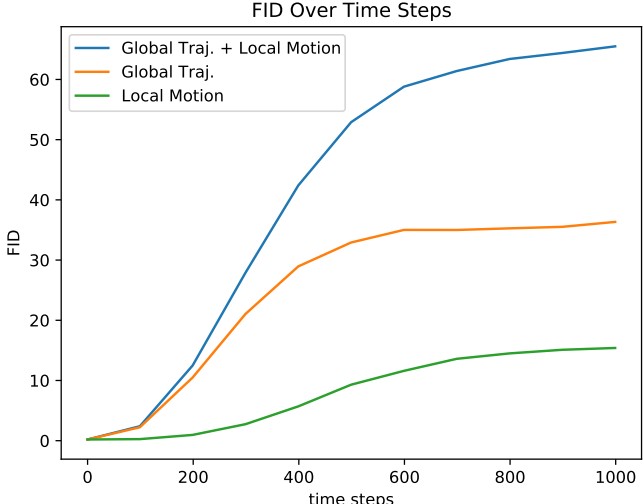

Figure C.1: **Change of FID for different noising modes and diffusion steps.** Adding noise to the global trajectory has a more detrimental effect on the realism of interactions compared with adding noise to the local motion.

and the actor. The velocity interaction loss $L_I^v$ can be formulated as:

$$L_I^v = \frac{1}{|S'|} \sum_{(i,j,k) \in S'} \boldsymbol{W}_v[i,j,k] \|\tilde{\boldsymbol{M}}_v[i,j,k] - \boldsymbol{M}_v[i,j,k]\|_2^2, \tag{11}$$

where $S' = \{(i,j,k) | \boldsymbol{D}_p[i,j,k] \le c, \ k < T\}$ is a set of index pairs and $\boldsymbol{W}_v = \sigma(\tilde{\boldsymbol{D}}_p) + \sigma(\boldsymbol{D}_p)$ is the weighted term. The definition of $\tilde{\boldsymbol{D}}_p$, $\boldsymbol{D}_p$, and $\sigma$ are consistent with those in the main paper.

**Formulation of Kinematic Loss $L_K$.** As shown in Sec. 3.3 of the main paper and building upon (Tevet et al., 2023; Liang et al., 2024; Ghosh et al., 2024), we use a kinematic loss term, $L_K$, to prevent artifacts like foot sliding or jittering. Moreover, we aim to utilize the kinematic loss $L_K$ to make our model focus more on the generation of global trajectory. This focus is crucial because, despite the greater importance of global trajectory compared with local motion, the motion representation allocates only 4 values for the global trajectory versus 259 for the local motion. As shown in GMD (Karunratanakul et al., 2023), such a disparity could lead the model to prioritize local motion generation. Therefore, we want to use the kinematic loss $L_K$ to eliminate such a bias.

Specifically, $L_K$ consists of 4 subterms, which can be formualted as:

$$L_K = \lambda_{\text{foot}} L_K^{\text{foot}} + \lambda_{\text{vel}} L_K^{\text{vel}} + \lambda_{\text{rot}} L_K^{\text{rot}} + \lambda_{\text{traj}} L_K^{\text{traj}}. \tag{12}$$

Here, $L_K^{\text{foot}}$, $L_K^{\text{vel}}$, $L_K^{\text{rot}}$, and $L_K^{\text{traj}}$ correspond to the foot skating loss, velocity loss, global rotation loss, and global position loss, respectively. The coefficients $\lambda_{\text{foot}}$, $\lambda_{\text{vel}}$, $\lambda_{\text{rot}}$, and $\lambda_{\text{traj}}$ denote the weights assigned to these four loss terms. To compute these losses, we first compute joint positions $\tilde{\boldsymbol{P}}_x, \boldsymbol{P}_x \in \mathbb{R}^{J \times 3T}$ and joint velocities $\tilde{\boldsymbol{V}}_x, \boldsymbol{V}_x \in \mathbb{R}^{J \times 3(T-1)}$ of the generated reaction and ground truth reaction. We further use $\tilde{\boldsymbol{R}}_x, \boldsymbol{R}_x \in \mathbb{R}^T$ to denote the global rotation of the reactor along the y-axis. For clarity, we omit the subscript $x$ in subsequent formulations.

To compute the foot skating loss $L_K^{\text{foot}}$, we first calculate $\tilde{\boldsymbol{H}} \in \mathbb{R}^{J \times (T-1)}$, which signifies the height of each joint across the previous $T-1$ frames. The formulation of $L_K^{\text{foot}}$ is then articulated as follows:

$$C[i] = I(\|\tilde{\boldsymbol{V}}[i]\|_2 \le \gamma_v) * I(\tilde{\boldsymbol{H}}[i] \le \gamma_h) \tag{13}$$

$$L_K^{\text{foot}} = \frac{1}{\sum_{i \in \text{FootJoints}} C[i]} \sum_{i \in \text{FootJoints}} \|\tilde{\boldsymbol{V}}[i]\|_2^2 * C[i]. \tag{14}$$

Here, $\gamma_v$ and $\gamma_h$ serve as thresholds for calculating $C$, where $C \in \{0,1\}^{J \times (T-1)}$ indicates the contact between each joint and the ground in each frame. FootJoints $\subset \{1, \cdots, J\}$ represents the subset of indices corresponding to foot joints.

We use similar equations to compute velocity loss $L_{\mathrm{K}}^{\mathrm{vel}}$, global rotation loss $L_{\mathrm{K}}^{\mathrm{rot}}$, and global position loss $L_{\mathrm{K}}^{\mathrm{traj}}$, which are expressed as follows:

$$L_{\mathrm{K}}^{\mathrm{vel}} = \frac{1}{J * (T-1)} \sum_i \|\tilde{\boldsymbol{V}}[i] - \boldsymbol{V}[i]\|_2^2 \tag{15}$$

$$L_{\mathrm{K}}^{\mathrm{rot}} = \frac{1}{T} \|\tilde{\boldsymbol{R}} - \boldsymbol{R}\|_2^2 \tag{16}$$

$$L_{\mathrm{K}}^{\mathrm{traj}} = \frac{1}{T} \|\tilde{\boldsymbol{P}}[\mathrm{root}] - \boldsymbol{P}[\mathrm{root}]\|_2^2, \tag{17}$$

where 'root' denotes the index of the root joint of the reactor.

**Experimental Setup of InterGen.** Originally designed for text-driven human interaction generation, InterGen (Liang et al., 2024) processes a text prompt $w$ to generate interactions $\boldsymbol{z} = [\boldsymbol{x}, \boldsymbol{y}]$ between two people with respect to $w$. However, it is *not* directly applicable to our task of text-driven human reaction generation. To adapt InterGen for this new task, we integrate an inpainting mechanism into the inference process of InterGen, which is similar to the method described in Sec. 3.4 of the main paper. At each time step $t$ of the denoising process of InterGen, after estimating the clean interaction $\tilde{\boldsymbol{z}}_0 = [\tilde{\boldsymbol{x}}_0, \tilde{\boldsymbol{y}}_0]$, we embed the known actor's motion into $\tilde{\boldsymbol{z}}_0$ to obtain $\hat{\boldsymbol{z}}_0$. This operation can be expressed as:

$$\hat{\boldsymbol{z}}_0 = [\boldsymbol{1}, \boldsymbol{0}] \odot \tilde{\boldsymbol{z}}_0 + [\boldsymbol{0}, \boldsymbol{1}] \odot [\boldsymbol{0}, \boldsymbol{y}] = [\tilde{\boldsymbol{x}}_0, \boldsymbol{y}]. \tag{18}$$

Here, $\odot$ denotes the Hadamard product. The modified result, $\hat{\boldsymbol{z}}_0 = [\hat{\boldsymbol{x}}_0, \hat{\boldsymbol{y}}_0]$, is subsequently utilized to calculate $\boldsymbol{\mu}_t$ and to sample $\boldsymbol{z}_{t-1}$ from $\mathcal{N}(\boldsymbol{\mu}_t, \boldsymbol{\Sigma}_t)$. By employing this inpainting mechanism, we continuously integrate the known actor's motion $\boldsymbol{y}$ throughout the denoising process, guaranteeing that the resulting interaction $\boldsymbol{z} = [\boldsymbol{x}, \boldsymbol{y}]$ accurately conforms to $\boldsymbol{y}$. Consequently, $\tilde{\boldsymbol{x}}_0$ in the ultimate denoising outcome $\boldsymbol{z}_0 = [\boldsymbol{x}_0, \boldsymbol{y}_0]$ represents the reaction generated with respect to both the textual prompt $w$ and the actor's motion $\boldsymbol{y}$.

Through communication with the authors of InterGen (Liang et al., 2024), we discovered that the publicly released checkpoint (Liang et al., 2023) of InterGen was trained using both the training and test sets to produce best demonstrations. Therefore, for a fair comparison, we train InterGen from scratch using their codebase, strictly following the experimental setup presented in their paper.

**Experimental Setup of MDM.** We adapted the official code of MDM to suit the text-driven reaction generation task. Specifically, by concatenating the action and reaction features before feeding them into the model, we enable the model to be aware of the interaction between two people instead of focusing on just one person. We experimented with two backbones for MDM: a transformer encoder-only backbone and a GRU backbone. For the transformer encoder-only backbone, we utilized N=8 blocks, each with a latent dimension of 1,024, and equipped each attention layer with 8 heads. For the GRU backbone, we set N=8 GRU layers with a latent dimension of 1,024. Both models were trained for 2,000 epochs using the AdamW optimizer, consistent with the training settings of MoReact.

**Detailed Model Configurations.** In the transformer-style architecture of our full-body motion diffusion model, we utilize N=8 blocks, each with a latent dimension of 1,024, and we equip each attention layer with 8 heads, consistent with the setup in InterGen (Liang et al., 2024). Before inputting the noised reaction vector $\boldsymbol{x}_t$ into the transformer layers, we use a linear layer to adjust its dimension to match the transformer's input dimension. Similarly, the output from the transformer layers is processed by another linear layer to match the motion feature's dimension. For text processing, we utilize a frozen `CLIP-ViT-L-14` model to encode the text prompt into text features for cross-attention. Moreover, following InterGen, we extract the most salient text feature embedding, combine it with the diffusion timestep feature, and employ this composite feature within the adaptive layer norms of the transformer blocks. To encode the actor's motion $\boldsymbol{y}$, a transformer encoder layer comprising 2 blocks, a latent dimension of 1,024, and 8 heads per attention

Two individuals are seated on chairs, the first individual is looking at the phone, while the other suggests that the former should look ahead.

The other person forcefully raises their right hand and slaps the left cheek of one person, causing them to collapse to the ground. The other person stands by the side afterwards.

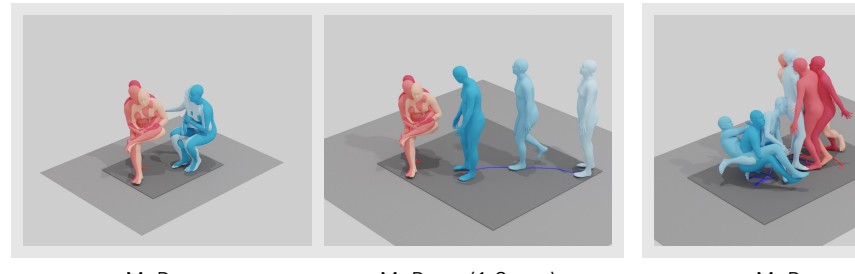
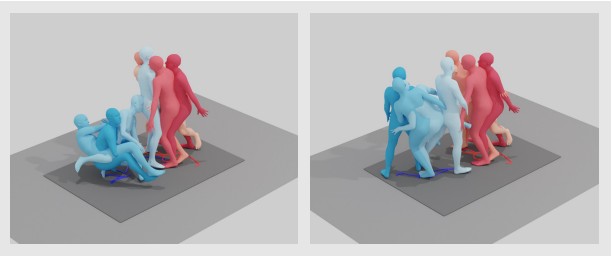

| MoReact | MoReact (1-Stage) | MoReact | MoReact (1-Stage) |

Figure E.1: **Ablation study** on the design choice within MoReact.

layer is utilized prior to incorporating $\boldsymbol{y}$ for cross-attention. Except for the absence of a cross-attention layer, the architecture of the trajectory diffusion model mirrors that of the full-body diffusion model.

During training, we use a 1,000-step diffusion process and adopt a classifier-free technique (Ho & Salimans, 2022) that randomly masks 10% of the text conditions, 10% of the actor's motion conditions, and 10% of the global trajectory condition independently. During inference, we use the DDIM (Song et al., 2020) sampling strategy with 50 time steps and $\eta = 0$, and set the classifier-free guidance coefficient $s = 3.5$. For the hyperparameters used in the training of the revised model, we set $(\lambda_R, \lambda_K, \lambda_I, \lambda_K^{\text{foot}}, \lambda_K^{\text{vel}}, \lambda_K^{\text{rot}}, \lambda_K^{\text{traj}}, L_I^p, L_I^v)$ to $(7.0, 1.0, 1.0, 300.0, 110.0, 1.5, 10, 5.0, 25.0)$, respectively. In addition, we set the threshold $\bar{t}$ for applying the kinematic loss $L_K$ and the interaction loss $L_I$ as 700.

**Details for experiments on CHI3D dataset.** To demonstrate the generalization ability of MoReact, we adapted it to suit the action-driven reaction generation task and evaluated it on the CHI3D (Fieraru et al., 2020) dataset. Specifically, instead of using CLIP to extract features from text as in the text-driven reaction generation task, we employed a learnable action embedding to encode the action features. Additionally, compared to the architecture shown in Fig. 2(b) of the main paper, we eliminated the cross-attention layer that fuses the textual features into the denoising process. We reduced the latent dimension to 512 and the batch size to 16. The model was trained for 1,000 epochs using the AdamW optimizer. We also made corresponding adjustments to the baseline MDM model (reducing the latent dimension, adjusting the batch size, and training settings) to ensure a fair comparison. We follow the official implementation of ST-GCN (Yan et al., 2018) to build our evaluator, an interaction classifier trained on CHI3D.

# E  Additional Ablation Studies

**Two-Stage vs. Single-Stage.** Beyond the quantitative analysis of the design choice in Sec. 4.4 of the main paper, we present some visual results generated by both the two-stage and single-stage frameworks. As illustrated in Fig. E.1 and supplementary video, our two-stage framework generates more natural and text-aligned reactions compared to the single-stage baseline, validating the effectiveness of our two-stage approach.

**Predicted Term of Trajectory Diffusion Model.** As mentioned in Sec. 3.2 of the main paper, diffusion models can employ two kinds of strategies during the denoising process to derive $\boldsymbol{x}_{t-1}$ from the noised data $\boldsymbol{x}_t$: predicting the noise $\boldsymbol{\epsilon}$, or predicting the clean data $\boldsymbol{x}_0$. Here, we conduct experiments to determine which approach is more effective for the trajectory diffusion model. The results, displayed in Table E.1, indicate that the variant focusing on noise prediction $\boldsymbol{\epsilon}$ outperforms, aligning with the conclusions drawn by GMD (Karunratanakul et al., 2023).

Table E.1: Ablation studies on predicted term of trajectory diffusion model. The trajectory model that predicts $\epsilon$ achieves better performance in R-precision, FID and Multi-Modality Distance.

| Methods | Traj. Model | 3-Precision$^\uparrow$ | FID$^\downarrow$ | MM Dist$^\downarrow$ | Diversity$^\rightarrow$ |
|---|---|---|---|---|---|
| Real | - | $0.704^{\pm 0.005}$ | $0.206^{\pm 0.009}$ | $3.784^{\pm 0.001}$ | $7.799^{\pm 0.031}$ |
| MoReact | predict $\boldsymbol{x}_0$ | $0.568^{\pm 0.006}$ | $2.959^{\pm 0.030}$ | $3.826^{\pm 0.001}$ | $\mathbf{7.808}^{\pm 0.030}$ |
| MoReact | predict $\epsilon$ | $\mathbf{0.615}^{\pm 0.007}$ | $\mathbf{2.412}^{\pm 0.050}$ | $\mathbf{3.813}^{\pm 0.002}$ | $7.775^{\pm 0.046}$ |

**Interaction Loss.** While some existing work also employed interaction loss to facilitate interaction generation, their implementations differ from ours in some important aspects. For example, ReMoS (Ghosh et al., 2024) only considers corresponding joints of the interacting individuals in its interaction loss, thus failing to capture diverse joint interaction patterns present in real-world scenarios. InterGen (Liang et al., 2024), on the other hand, does not incorporate a weighting mechanism, preventing it from effectively penalizing unrealistic close interactions or appropriately de-emphasizing irrelevant distant ones. In contrast, our interaction loss introduces a novel weighting mechanism that dynamically adjusts the importance of joint pairs based on both ground-truth and generated interactions, thereby enabling more realistic reaction generation. Additionally, we re-implemented the interaction losses employed by InterGen and ReMoS within MoReact and conducted quantitative comparisons with our method. As demonstrated in Table E.2, our approach consistently achieves superior performance in terms of R-precision, FID, and MM Dist, highlighting the effectiveness of our weighted interaction loss.

Table E.2: Quantitative comparison of different interaction loss designs. Our weighted interaction loss consistently outperforms InterGen (Liang et al., 2024) and ReMoS (Ghosh et al., 2024) losses on R-precision, FID, and MM Dist, demonstrating its superior effectiveness in generating realistic reactions.

| Methods | 3-Precision$^\uparrow$ | FID$^\downarrow$ | MM Dist$^\downarrow$ | Diversity$^\rightarrow$ |
|---|---|---|---|---|
| Real | $0.704^{\pm 0.005}$ | $0.206^{\pm 0.009}$ | $3.784^{\pm 0.001}$ | $7.799^{\pm 0.031}$ |
| InterGen (Liang et al., 2024) Loss | $0.596^{\pm 0.009}$ | $3.436^{\pm 0.075}$ | $3.826^{\pm 0.002}$ | $7.887^{\pm 0.039}$ |
| ReMoS (Ghosh et al., 2024) Loss | $0.608^{\pm 0.007}$ | $2.817^{\pm 0.070}$ | $3.819^{\pm 0.002}$ | $\mathbf{7.792}^{\pm 0.035}$ |
| MoReact | $\mathbf{0.615}^{\pm 0.007}$ | $\mathbf{2.412}^{\pm 0.050}$ | $\mathbf{3.813}^{\pm 0.002}$ | $7.775^{\pm 0.046}$ |

# F Limitations and Social Impacts

**Limitations and Future Work.** MoReact is designed to generate reactions by considering both textual descriptions and the motion of another individual. Future research will aim to generalize our method to broader contexts, for example, generating reactions based on text and the motions of multiple people.

**Potential Social Impact.** We recognize the potential application of reaction synthesis in military training contexts. With our model, the military might generate a virtual soldier who can dodge and counteract in response to a real soldier's movements, thereby simulating authentic battlefield scenarios to train soldiers.

