# OpenReview forum: "MoReact: Generating Reactive Motion from Textual Descriptions"
_TMLR — Accepted by TMLR_

### Review · Reviewer_r2bf · 2025-06-07

**Summary Of Contributions:**

This paper focuses on the task of reactive action generation given text prompts and actor actions. The authors propose a two-stage approach: first generating trajectories, then generating responsive actions based on the trajectories. The authors demonstrate the effectiveness of the approach both quantitatively and qualitatively compared to baseline methods and present an extensive ablation studies on design choices.

**Audience:**

Yes

**Claims And Evidence:**

No

**Requested Changes:**

Please see my detailed comments in the [Weaknesses] section above. I think the paper needs significant revision to meet TMLR standards.

**Strengths And Weaknesses:**

Strengths
- The authors propose an interesting new task: reactive motion generation.
- The authors proposed some novel training losses - kinematic loss and interaction loss given the task.
- The paper is well written and easy to understand

Weaknesses
- I am confused about the ablation study in Table 2, it seems that the method without kinematic loss and with interaction loss show better overall performance in all metrics. Please explain the underlying reason. I am not completely convinced that the kinematic loss is effective at this point.
- Although the proposed task is novel, I feel it is much simpler than the two-body motion estimation proposed in InterGen, because in this task we additionally have the known motion of one human body with paired text prompt, which greatly simplifies the task. Therefore, even if you provide the actor's motion information in inference, I don't think it is a fair comparison with InterGen, and I am not surprised that the proposed model is better than other methods.
- The figures are a bit misleading because it is shown with the SMPL model, while the full method does not use SMPL in fact, and the interaction loss does not consider the contact and disjointness between the two SMPL meshes.
- The proposed two-stage prediction method is not novel idea in motion generation.

---

> ### Author Response · Authors · 2025-06-27
> **Response to Reviewer r2bf**
>
> We sincerely thank the reviewer for the valuable feedback. We address each point below in the order presented.
>
> 1. - We appreciate the reviewer’s observation. While the variant without kinematic loss and with interaction loss performs well on some metrics, we emphasize that our full model achieves the **best FID**, which we regard as the **most critical metric** for evaluating the realism and overall quality of the generated reactions.
>
>     - Please note, in our task, **R-Precision** and **MMDist** do not fully reflect motion quality. We observe that models like InterGen and MDM-GRU often achieve relatively good R-Precision and MMDist scores while producing unrealistic motions that contain artifacts such as body penetrations or physically implausible poses (as you can see in the Figure 3 or supplementary video). This happens because these metrics are based on text-motion alignment, and the evaluator model does not explicitly penalize poor motion realism. For example, in a "fighting" scenario, a motion with clear fighting gestures but unnatural self-intersections may still score well in alignment.
>
>     - Specifically, our kinematic loss mitigates motion artifacts such as foot sliding, thereby improving the **visual realism** of the generated motion—as evidenced by the improved FID score. It is reasonable that our model shows slightly lower R-Precision compared to the baseline, which directly optimizes text-motion alignment via reconstruction loss.
>
> 2. - We respectfully disagree with the notion that our task is simpler than InterGen. Providing the actor’s motion as input does not simplify the problem; instead, it introduces a **consistency constraint**, as the generated reaction must be well-aligned with the observed motion—an aspect not addressed by InterGen, which focuses on generating motion sequences for both characters jointly. To successfully perform our task, the model must generate a **semantically appropriate**, **physically plausible**, and **temporally coherent** reaction that aligns with both the actor’s motion and the intent expressed in the text—making the problem highly challenging.
>
>     - Moreover, the inherently complex nature of human reactions further increases the difficulty of this task. The same actor motion or text can lead to vastly different reactions depending on the context. Nevertheless, as illustrated in Figure 4, our model is able to capture this complexity and demonstrates **controllability** with respect to both the actor’s motion and the text input —a level of controllability that is difficult to achieve for existing models not explicitly designed for reaction generation.
>
>     - To ensure a fair comparison with InterGen, we also implemented a text-only baseline (InterGen* in Table 1) for InterGen. The results show that even without the additional requirement of aligning with the actor’s motion, InterGen still produces lower-quality motions compared to our method.
>
> 3. We respectfully clarify that the use of SMPL in our figures is solely for visualization purposes, following a **widely adopted convention** in the human motion generation community. Many influential works—such as MDM[A], MotionDiffuse[B], and MotionGPT[C]—do not incorporate SMPL into their model architecture or training objectives, but still use it to render motion for clearer presentation and comparison.
> 4. We thank the reviewer for the comment. While two-stage architectures have been explored in the broader motion generation literature, our work is the first to introduce such a design specifically for the **reaction generation task**, which presents unique challenges—namely, producing reactions that are semantically aligned with the text while also being physically and temporally consistent with the actor’s motion. Our two-stage framework is built on the key insight that *the global trajectory serves as the foundation for both local motion and interaction realism,* which is carefully validated in Appendix Section C and Figure 1(b)(c). Furthermore, its effectiveness is strongly supported by our experiments (Table 2, Figure E.1). We believe this constitutes a meaningful and task-specific contribution, going beyond superficial architectural reuse.
>
> [A] "MDM: Human Motion Diffusion Model." The Eleventh International Conference on Learning Representations. 2023.
>
> [B] "Motiondiffuse: Text-driven human motion generation with diffusion model."  IEEE transactions on pattern analysis and machine intelligence. 2024.
>
> [C] "MotionGPT: Human Motion as a Foreign Language." Advances in Neural Information Processing Systems. 2023.

---

### Review · Reviewer_cwLr · 2025-06-13

**Summary Of Contributions:**

This paper proposes a framework for generating reactive motions based on an actor's movements and textual descriptions. The method decouples motion generation into global trajectory planning and local full-body articulation, adopting a two-stage design for reaction synthesis. To enhance the realism of the generated motions, the authors introduce an interaction loss that constrains relative spatial relationships between interacting agents. Experimental results demonstrate the effectiveness of the proposed approach, showing improved motion quality and interaction plausibility.

**Audience:**

Yes

**Broader Impact Concerns:**

N.A

**Claims And Evidence:**

Yes

**Requested Changes:**

- The paper should include comparisons with additional baseline methods to better demonstrate the effectiveness of the proposed approach.
- The paper would benefit from a clearer discussion of how the proposed loss function differs from those used in previous work.

**Strengths And Weaknesses:**

Strengths
- This paper decomposes reaction generation into global trajectory generation and full-body motion generation, reducing the complexity of the task.
- The motivation is clear. The writing is easy to follow.

Weaknesses
- How is the quality of the global trajectory ensured? An inaccurate global trajectory may severely impact the quality of the subsequent motion generation.
- The interaction loss actually penalizes the relative joint positions between two characters, a concept that has already been introduced in previous works [A, B]. Therefore, the differences between this work and those prior approaches should be clearly discussed.

     [A]  "Closely interactive human reconstruction with proxemics and physics-guided adaption." Proceedings of the IEEE/CVF Conference on Computer Vision and Pattern Recognition. 2024.

     [B] "Intergen: Diffusion-based multi-human motion generation under complex interactions." International Journal of Computer Vision 132.9 (2024): 3463-3483.
- While the diffusion architectures are effective, they are largely based on prior works and do not introduce significant innovations specific to the challenges of reaction generation.
- Since InterGen and MDM are not specifically designed for reaction generation tasks, the proposed method should also be compared with baseline approaches that are explicitly tailored for this problem. Regennet can also input action label.

     [A] "Regennet: Towards human action-reaction synthesis." Proceedings of the IEEE/CVF Conference on Computer Vision and Pattern Recognition. 2024.

     [B] "Remos: 3d motion-conditioned reaction synthesis for two-person interactions." European Conference on Computer Vision. Cham: Springer Nature Switzerland, 2024.
- "black mesh" in the caption of Fig.1 might be a typo.

---

> ### Author Response · Authors · 2025-06-27
> **Response to Reviewer cwLr**
>
> We greatly appreciate the reviewer’s comments, which help us improve the clarity and impact of our work.  Below we address every concern in the order raised.
>
> 1. - We thank the reviewer for highlighting this important point. While we agree that an entirely inaccurate global trajectory (e.g., crossing paths in a non-interacting way) can lead to unrealistic reactions, our two-stage framework is designed to be **robust to moderate variations** in trajectory quality. In fact, the second-stage full-body motion generator is able to adapt flexibly to the trajectory as long as the **overall trend is semantically correct**.
>
>     - For example, in a “approaching for a handshake” scenario, as long as the two agents’ trajectories move from far apart to close proximity, the second module can generate a realistic reaching motion—regardless of small variations in step length or final position. If the trajectory ends earlier, the model produces a shorter walk; if it ends farther away from the actor, the model naturally extends the hand further.
>
>     - We emphasize that our trajectory diffusion module is designed to capture the **key semantic trends** of motion (e.g., approach, avoidance, alignment), which are sufficient to guide plausible full-body reactions. As discussed in Section 3, this is not a particularly difficult task, and our module is well-suited for this purpose. We further analyze the impact of different trajectory diffusion configurations on generation quality in Appendix Table E.1, which supports the soundness of our design choices.
>
> 2. - We appreciate the reviewer’s insightful comments. While our interaction loss shares the general idea of modeling relative joint positions with prior works [A, B], there are **two key differences** that distinguish our formulation:
>       1. **Distance-aware weighting mechanism.** Prior works treat all joint pairs equally, assigning uniform importance regardless of spatial proximity. In contrast, our method introduces a distance-based weighting scheme, which assigns higher importance to close joint pairs—such as hands during a handshake—and reduces the influence of distant or irrelevant joints. This design enables more accurate modeling of contact-sensitive interactions and helps penalize implausible close-contact motions more effectively.
>       2. **Inclusion of both position and velocity terms.** Our interaction loss goes beyond static joint displacement by also considering relative joint velocities. This dynamic term encourages temporal coherence in interactions.
>
>     - We will revise the paper to better highlight these distinctions and clarify how our formulation improves upon prior work.
>
> 3. - We thank the reviewer for the comment. While our diffusion-based backbone draws inspiration from prior works, our method introduces substantial task-specific innovations tailored to the challenges of reaction generation:
>       1. We propose a novel two-stage generation framework that decouples global trajectory synthesis from full-body motion generation;
>       2. We propose a weighted interaction loss that differs from previous works by explicitly incorporating a **distance-aware weighting scheme** and modeling both **relative joint positions and velocities**.
>
>    - We believe these contributions go beyond architectural re-use and represent meaningful innovations for this emerging task.
>
> 4. - We thank the reviewer for suggesting additional baselines. ReGenNet and ReMoS are indeed relevant works targeting reaction generation. While we have not conducted full model-level comparisons due to time and resource constraints, we have carefully studied their **interaction loss formulations** and compared them under our framework. Specifically:
>       1. We implemented the interaction loss described in ReMoS within our model and reported results in Appendix Table E.2. Our proposed interaction loss leads to better results, as ReMoS only considers corresponding joints between interacting characters, which limits its ability to model the rich and diverse joint interactions observed in real-world scenarios.
>       2. We also found that ReGenNet’s interaction loss actually degenerates to a simple reconstruction loss, offering limited benefit for learning reactive behavior. This is an example of one of the components of its interaction loss:
>
>     $$\theta^{x\rightarrow y}=FK(\theta^y)-FK(\theta^x), L_{inter}^\theta=||\theta^{x_0\rightarrow y}-\theta^{\hat x_0\rightarrow y}||_2^2=||FK(\theta^{x_0})-FK(\theta^{\hat x_0})||_2^2$$
>
>     - Due to the lack of public implementation for ReMoS and the engineering overhead required to adapt Regennet, we were unable to complete full experimental comparisons now. If the paper is accepted, we would be happy to include these additional comparisons in future versions or extended work.
>
> 5. We thank the reviewer for pointing this out. It should be “blue mesh”. We will revise the caption to clarify this in the camera-ready version.

---

### Review · Reviewer_PFaC · 2025-06-15

**Summary Of Contributions:**

This paper propose a new reactive motion generation framework conditioned on text descriptions, which also introduces a new workstream which has been under-explored before. The proposed method consists of two modules, a trajectory diffusion module and a full-body motion diffusion module, and authors have been exploring global motion and local motions during the modeling human motion generation. Authors also conduct experiments to validate the proposed method as well as losses used during the model training.

**Audience:**

Yes

**Broader Impact Concerns:**

Authors have discussed the social impacts in the appendix.

**Claims And Evidence:**

No

**Requested Changes:**

I have listed some of my main concerns in the weakness section. My main concern lies in evaluations since this paper proposes a new task, reactive motion generation with text descriptions. I would expect more extensive evaluations to support this work and benefit following work.

**Strengths And Weaknesses:**

Strengths:

Overall, the paper defines the problem in a clear way, provides extensive literature review and explain the proposed method as well as experiments in a clear way. Training details have been explained for reproduction and both qualitative and quantitative results have been shown and discussed.

Weaknesses:

I have listed some of my concerns which are expected to be further clarified by authors:

1. In the second last paragraph of page 4, I recommend authors further clarify how the global coordinate and local coordinate are defined? a good illustration figure would be helpful.

2. Overall, it is not clear to me whether the method is image-based or video-based? Authors can further explain this.

3. Since authors mentioned that "_Given that text-driven human reaction generation is a novel task, there is no existing work or
code publicly available that directly serves as a baseline for comparison._". My expectations are this work can (1) setup a baseline/benchmark for following tasks; (2) experiments and comparisons are to be extensive on a varieties of datasets and environments, indoor, outdoor, single/multiple person, etc. Given the current experimental results (Table 1, Table 2), this work (evaluation breadth and evaluation depth) is not strong enough.

4. I wonder how quality of text descriptions can affect generation results?

5. I did not see any failure cases.

---

> ### Author Response · Authors · 2025-06-27
> **Response to Reviewer PFaC**
>
> We sincerely thank the reviewer for the detailed and constructive feedback. Below we address every concern in the order raised.
>
> 1. We thank the reviewer for this helpful comment. In the camera-ready version, we will add an illustration figure to better explain the concepts of global trajectory and local pose information. Specifically, the global trajectory represents the person’s movement path in the world coordinate system, encoding the root’s horizontal position, height, and facing orientation. It reflects **where** the person is moving in the scene over time. In contrast, local pose information describes the person’s detailed body articulation in their own local coordinate system, including joint positions, velocities, rotations, and foot contact states. It reflects **how** the person’s body moves, independent of their global position.
> 2. We thank the reviewer for raising this point. To clarify, our method is **motion-based**, not image- or video-based. Specifically, both the actor’s motion and the generated reactor’s motion are represented as sequences of 3D poses, parameterized as global trajectory and local pose information for each frame (see Task Definition, Sec. 3). Our model operates entirely in this structured motion representation space, without relying on image or video inputs or outputs. Visualization of the results is performed by rendering the generated motion sequences as animated meshes for illustration purposes.
> 3. - We sincerely thank the reviewer for this valuable perspective. We fully agree that establishing strong baselines and benchmarks is crucial for advancing future research in this direction. In this work, however, our focus is on the novel and under-explored task of **text-driven reactive motion generation**. This is clearly different from broader single-person, multi-person (>2), or scene-conditioned (indoor/outdoor) motion generation tasks, which often involve very different representations and modeling challenges. Expanding to those domains is beyond the intended scope of this work.
>
>     - Within this defined scope, we have conducted comprehensive evaluations:
>
>       1. We covered two different datasets (InterHuman, CHI3D), which already vary in interaction types and settings. To the best of our knowledge, these two datasets are the only publicly available labeled datasets describing high-quality human interactions at the time of paper submission;
>       2. We adapted multiple strong baselines (MDM, InterGen) to the reaction generation task;
>       3. We provided extensive quantitative, qualitative, and ablation studies (Tables 1 & 2 & E.1 & E.2, Figure 3 & 4 & E.1, Supplementary Video), validating both model components and generation quality.
>
>     - We believe this level of breadth and depth is appropriate and sufficient for the scope of this initial benchmark effort on text-drivenreaction generation. If the paper is accepted, we plan to release both code to enable the community to further extend the evaluation breadth in future work.
>
> 4. We thank the reviewer for this question. In practice, it is challenging to formally define “text quality.” Our model is trained on the textual descriptions provided in the InterHuman dataset. We observe that if the input text follows a similar style or distribution to this training data—even if it contains minor typos or grammatical errors—the model is still able to generate reactions that are semantically aligned with the text. Otherwise, if the input text deviates significantly from the training distribution, the generated motion may not align well with the intended meaning of the text.
> 5. We thank the reviewer for pointing this out. Some failure cases—such as foot skating artifacts—can be observed in the supplementary video. We will further expand the discussion of failure cases and potential limitations in the revised version and future work.

---

### Decision · Action_Editor_wju2 · 2025-08-05

**Recommendation:** Accept as is

**Audience:**

Yes

**Audience Explanation:**

The work is in the interest of ML community.

**Claims And Evidence:**

Yes

**Claims Explanation:**

The authors propose a method for generating reactive motions for textual descriptions. The method is sound , and the inclusion of relationship loss is important.